Exploring reported population differences in Norway lobster (Nephrops norvegicus) in the Pomo Pits region of the Adriatic Sea using genome-wide markers

Jenkins Tom L. t.l.jenkins@exeter.ac.uk tom.l.jenkins@outlook.com 1
Martinelli Michela 2
Ellis Charlie D. 1
Stevens Jamie R. j.r.stevens@exeter.ac.uk j.r.stevens@ex.ac.uk 1
1 Department of Biosciences, Faculty of Health and Life Sciences, University of Exeter , Exeter , Devon , United Kingdom
2 National Research Council, Institute for Marine Biological Resources and Biotechnologies (CNR IRBIM) , Ancona , Italy
Johnson Magnus
Electronic publication date: 2024 Oct 21
Publication date: 2024
Volume: 12
Electronic Location ID: e17852
Received 2024 Apr 17; Accepted 2024 Jul 11
Copyright: ©2024 Jenkins et al.
Copyright year: 2024
Copyright holder: Jenkins et al.
License: This is an open access article distributed under the terms of the Creative Commons Attribution License, which permits unrestricted use, distribution, reproduction and adaptation in any medium and for any purpose provided that it is properly attributed. For attribution, the original author(s), title, publication source (PeerJ) and either DOI or URL of the article must be cited.
License URL: https://creativecommons.org/licenses/by/4.0/

Keywords: Dublin Bay prawn, Mediterranean, Nephrops norvegicus, Pomo/Jabuka Pit, Gene flow, Fisheries management, Population genetic structure, Single nucleotide polymorphisms, Adriatic Sea

Funding: The European Union Joint Research Centre, Water and Marine Resources Unit No. 722111 This research was supported by a grant (Genetic analysis of Nephrops norvegicus in the Adriatic Sea, No. 722111) from the European Union Joint Research Centre, Water and Marine Resources Unit. The funders had no role in study design, data collection and analysis, decision to publish, or preparation of the manuscript.

==============================
The Norway lobster (Nephrops norvegicus) is one of the most important decapod crustacean seafood species in the Adriatic Sea. Previous research has identified significant differences in growth rates and maturation timing of Nephrops in the Pomo/Jabuka Pits area compared to other subpopulations in Adriatic fishing grounds. Here, we use 1,623 genome-wide single nucleotide polymorphisms (SNPs) to investigate whether the Pomo Pits subpopulation is genetically different from other sites in the Adriatic and neighbouring seas. We found no genetic differentiation among all sampled Adriatic sites, suggesting high gene flow between Pomo Pits Nephrops and those of surrounding areas. We also found genetic homogeneity between the Adriatic sites and single-site samples from the Aegean and Tyrrhenian Seas. However, we detected distinct genetic differentiation between all Mediterranean sites and an Atlantic site in western Scotland, which provides evidence for a phylogenetic break between the Atlantic and the Mediterranean. Our results indicate that Pomo Pits Nephrops are not genetically different from others sampled in the Adriatic and that key biological parameters in Pomo Pits Nephrops could be driven by spatial variation in fishing pressure and/or environmental factors rather than geographic isolation.

Introduction

The Norway lobster (Nephrops norvegicus), hereafter Nephrops, is a benthic decapod crustacean found across the Mediterranean and the north-east Atlantic on the continental shelf and slope down to 800 m depth (Ungfors et al., 2013; Aguzzi et al., 2022). Nephrops construct and inhabit burrow systems used for shelter, usually on muddy seabed, and are not vulnerable to trawl capture when hiding in the substrate (Bell, Redant & Tuck, 2006; Aguzzi et al., 2021). The Nephrops fishery is extremely valuable to Europe, particularly in the Adriatic Sea and around the British Isles. In 2021, landings in the Adriatic Sea (∼537 tonnes) (FAO-GFCM, 2023) accounted for ∼30% of the total landings across the Mediterranean (∼1,846 tonnes) (FAO-GFCM, 2023), while landings in the United Kingdom and Ireland (∼38,505 tonnes) accounted for ∼71% of the total global landings (FAO-GFCM, 2023).

The Mediterranean is divided up into 27 geographical subareas (GSAs) established by the General Fisheries Commission for the Mediterranean (GFCM). The Adriatic Sea is split into two GSAs, GSA 17 and GSA 18 (Fig. 1A). In GSA 17, a deeper area characterised by three distinct benthic depressions known as the Pomo/Jabuka Pits, hereafter Pomo Pits, located between Italy and Croatia, is known to be a valuable spawning ground for Nephrops (Melaku Canu et al., 2021). Research has also shown significant differences in the biology of individuals from this area compared with Nephrops outside of Pomo Pits, such as smaller sized animals with slower average growth rates and individuals with smaller size at the onset of first maturity (SOM), which suggests the presence of a distinctive Nephrops subpopulation in Pomo Pits (Froglia & Gramitto, 1988; Angelini et al., 2020). In 2018, a Fisheries Restricted Area (FRA) was designated for Pomo Pits based on the GFCM/41/2017/3 recommendation (GFCM, 2017), which monitors Nephrops fishing in the area. Effort is restricted by spatial management and measures include limiting vessel numbers and permitted days, closed seasons and days-at-sea limits (Chiarini et al., 2022).

Figure 1 Study area, sampling information and carapace length measurements.

(A) Bathymetric map showing the sites sampled in the Adriatic Sea, the Aegean Sea and the Tyrrhenian Sea (Mediterranean), and the site sampled in the Firth of Clyde (Atlantic). The points in the Mediterranean are coloured by which geographical subarea (GSA) they are located in. Map data © 2024 Natural Earth. (B) Nephrops burrow (Image: © Grand Aquarium de Saint-Malo/CC BY-SA 3.0 DEED). (C) Carapace length variation at each site sampled (except 9I and Aeg) coloured by sex. The number of individuals used to visualise carapace length for each site are denoted by Nc in Table 1.

In this study, our main goal was to investigate whether the Nephrops Pomo Pits subpopulation, identified based on biological differences, also exhibits genetic differences compared with those outside of Pomo Pits in GSA 17, or with GSA 18 and neighbouring GSA stocks.

Materials and Methods

Tissue sampling and DNA extraction

Samples of adult Nephrops were collected from ten sites, including seven from the Adriatic Sea, one from the Aegean Sea, one from the Tyrrhenian Sea, and one from the Firth of Clyde in the northeast Atlantic (Table 1, Figs. 1A–1B). Samples from the Pomo Pits area were collected in November 2016 and October 2019 during the “ScamPo” experimental trawl survey, carried out yearly by CNR IRBIM of Ancona west of the Adriatic midline to monitor the effects of the management measures implemented since 2015. Samples from Ancona and Chioggia were collected in March and July 2019, respectively, by CNR IRBIM staff during biological sampling of commercial catches carried out within the European Data Collection Framework. Tissue samples were obtained by excising two pleopods or one pereiopod. All samples were placed in 95–100% ethanol and stored at 4 °C until DNA extraction. Genomic DNA was extracted using a salting-out protocol (Jenkins, Ellis & Stevens, 2018) and the quality of each extract was assessed on a 0.8% agarose gel. DNA purity was measured using a Nanodrop One and a Qubit 3.0 was used to quantify DNA concentration. The carapace length and sex of each animal sampled was also recorded (except for the Aegean and Tyrrhenian samples). We were able to obtain more data on carapace length and sex at each site (Nc in Table 1) because a surplus of Nephrops were processed during the sampling activity. These data (Nc) were used to visualise variation in carapace length at each site and by sex (Fig. 1C). To statistically assess differences in size between Pomo Pits (N = 172) and other Adriatic sites (N = 135), carapace length was modelled as a function of site (a binary variable describing whether an individual was sampled in or outside of Pomo Pits) and with sex (a binary variable describing an individual as male or female).

Table 1 Sampling information and genetic diversity statistics.

Area	Site	Code	N c	N g	Depth (m)	Year	Lat	Lon	P a	H o	H e	
Adriatic												
GSA 17	Off Ancona	Anc	34	16	70	2019	43.78	13.85	0	0.102	0.108	
GSA 17	Off Chioggia	Cgg	41	14	35-40	2019	45.18	13.24	0	0.096	0.097	
GSA 17	Pomo Pits	Pom1	132	9	216	2016	42.82	14.97	0	0.089	0.094	
GSA 17	Pomo Pits	Pom2	20	17	215-250	2019	42.85	14.74	1	0.108	0.111	
GSA 17	Pomo Pits	Pom3	20	10	170-180	2019	42.59	15.06	0	0.096	0.099	
GSA 17	17I	17I	30	5	93	2017	42.43	16.68	0	0.108	0.108	
GSA 18	18II	18II	30	5	92	2017	41.63	16.59	0	0.097	0.097	
Aegean												
GSA 22	Aegean Sea	Aeg	n/a	14	n/a	2019	40.17	23.54	0	0.108	0.113	
Tyrrhenian												
GSA 9	9I	9I	n/a	14	530	2017	41.40	12.20	0	0.102	0.105	
Atlantic	Firth of Clyde	Cly	32	8	50-75	2019	55.86	-4.90	9	0.095	0.096	
Notes.

GSA geographical subarea

Nc number of individuals sampled for carapace length measurement (total = 339)

Ng number of individuals genotyped (total = 112)

Pg number of private alleles

Ho mean observed heterozygosity

He mean unbiased expected heterozygosity

RAD sequencing and bioinformatics

DNA extracts for each sample were sent to Floragenex (Portland, Oregon, USA) for restriction site associated DNA sequencing (RAD-seq). RAD libraries were prepared for 112 samples using the SbfI restriction enzyme and sequenced on an Illumina sequencing platform using a 2 × 100 bp approach. Raw reads were trimmed using Fastp 0.20.1 (Chen et al., 2018) and further filtered using the process_radtags program from Stacks v2.53 (Catchen et al., 2013; Rochette, Rivera-Colón & Catchen, 2019). RAD loci were built using the Stacks de novo pipeline; default parameters were used for all modules, except for -m in ustacks which was set to 3. In addition, the following parameters were added to the populations command: (i) –min-samples-overall 0.75, (ii) –min-mac 5, and (iii) –write-single-snp. Using R v4.2.0 (R Core Team, 2022), the missingno() function from poppr v2.9.4 (Kamvar, Tabima & Grünwald, 2014) was used to remove any individuals with ≥ 30% missing genotypes. Functions from dartR v2.9.7 (Gruber et al., 2018; Mijangos et al., 2022) were used to filter out loci that: (i) departed from Hardy-Weinberg Equilibrium, (ii) were in linkage disequilibrium, or (iii) were monomorphic, had a minor allele count less than five, or had all missing genotypes in a single site. Lastly, OutFLANK v0.2 (Whitlock & Lotterhos, 2015) was used to identify any outlier loci.

Genetic diversity and population structure

The gl.report.heterozygosity() function from dartR was used to calculate observed and expected heterozygosity (Ho/ He). The gl.report.pa() function was used to calculate the number of private alleles per sampling site.

Genetic differentiation between sampling sites was assessed by calculating pairwise values of Fst (Weir & Cockerham, 1984) using the genet.dist() function from hierfstat v0.5-11 (Goudet & Jombart, 2022). Population structure was explored using two methods: (1) a principal components analysis (PCA), and (2) a genetic clustering analysis. Prior to analysis of population structure, missing data (NAs) were imputed with the gl.impute() function from dartR using the neighbour method. The PCA was then run using the glPca() function from adegenet v2.1.3 (Jombart & Ahmed, 2011). For genetic clustering, the optimal number of genetic clusters (K) to use was determined by running the snapclust.choose.k() function and visualising the Akaike information criteria (AICc) for each K. The find.clusters() function was executed to cluster individuals into K groups based on k-means clustering. The snapclust() function was then run on the data set for the chosen K. Snapclust uses maximum-likelihood estimations to assign individual membership probabilities to each K cluster (Beugin et al., 2018). The resulting membership probabilities to each K were mean averaged per site and visualised on a projected map of the study area. Isolation-by-distance (IBD) was tested by running a Mantel test on a dissimilarity matrix of pairwise genetic (Fst) and pairwise least-cost marine geographical distances (km) using the mantel.rtest() function from ade4 v1.7-22 (Thioulouse et al., 2018).

Effective population size

The gl.LDNe() function from dartR was used to calculate effective population size (Ne), which is a wrapper around the Ne Estimator v2.1 software (Do et al., 2014). The linkage disequilibrium method was run with random mating assumed, and jackknife 95% confidence intervals were computed. Given the homogeneity of sites from the Adriatic Sea (see Results), the Adriatic sites were divided into two groups, and different minor allele frequency thresholds were tested to compare results in case of inflationary effects by rare alleles. The first group comprised all three sites from Pomo Pits (Pom1, Pom2 and Pom3), and the second group comprised of the remaining sites from the Adriatic (17I, 18II, Anc, Cgg). The rationale here was to increase the sample size in each ‘population’ and assess whether there was a difference in Ne between these two groups, since small sample sizes are likely to severely bias the estimation of Ne (England et al., 2006).

Results

Carapace length analysis

Median carapace length in Pomo Pits was 28 mm for males (N = 95) and 24 mm for females (N = 77), while outside of Pomo Pits carapace length was 46 mm for males (N = 73) and 40 mm for females (N = 62). The model fitted to carapace length as a function of site and sex satisfied the assumptions of a linear regression (Supplementary Material S1). This model showed that, when sex is controlled for, individuals in Pomo Pits are on average 16.98 mm (41.5%) smaller than individuals outside Pomo Pits (P < 0.001). Additionally, the model showed that, when site is controlled for, males are on average 5.17 mm (12.6%) larger than females (P < 0.001).

RAD loci and genetic diversity

A total of 554 million reads were generated across 112 samples (mean = 3.78 million reads per sample). The de novo pipeline assembled 254,555 catalog RAD-tag loci, with an effective per-sample coverage mean of 4.7x (2.0x–35.7x). Filtering of individuals and loci produced a final data set of 98 individuals genotyped at 1,623 biallelic neutral SNPs; no loci were identified as outliers. At the Mediterranean sites, 0-1 private alleles were found at each site and mean observed heterozygosity was very similar across sites, ranging from 0.089−0.108. Mean unbiased expected heterozygosity values were close to observed heterozygosity, ranging from 0.094−0.113. In the single Atlantic site, the Firth of Clyde, nine private alleles were found, but heterozygosity was comparable to the Mediterranean sites.

Population genetic structure

Genetic differentiation was high between the Firth of Clyde and all Mediterranean sites, while differentiation amongst the Mediterranean sites was low or zero (Fig. 2A). The PCA revealed two distinct groups: individuals from the Firth of Clyde (Atlantic), and all other individuals from sites in the Mediterranean (Fig. 2B). This was also observed in the genetic clustering analysis, whereby K = 2 was the most likely number of ancestral populations (genetic clusters) (Fig. 2C), and this clearly showed Atlantic-Mediterranean separation into two distinct clusters (Fig. 2D). A PCA was run using only Mediterranean sites to check for any hierarchical structuring in the data (Fig. 3A). This revealed very little evidence for hierarchical structure among our sites sampled. In addition, there was little evidence for IBD across the range covered by our Mediterranean samples (Fig. 3B).

Figure 2 Population genetic structure results.

(A) Heatmap of pairwise Fst values for each site-site comparison. (B) Principal components analysis; each point represents the position an individual on axis 1 and 2 and colours correspond to sites located in the Atlantic (red) or the Mediterranean (blues). (C) The Akaike information criteria (AICc) scores for each K run using the snapclust algorithm; the plateau after K = 2 suggests that two is the most likely number of ancestral populations in the data set. (D) Projected map (ESPG: 3035) of the study area showing the membership proportion of individuals to each genetic cluster, averaged over each site; this map was produced using the mapmixture() function from mapmixture v1.1.0 (Jenkins, 2024). The north arrow points to the north pole. Map data ©2024 Natural Earth.

Figure 3 Hierarchical genetic structure results.

(A) Principal component analysis of Mediterranean sites only. (B) Scatter plot showing pairwise genetic (Fst) and geographic (km) distances between our sampling sites. The r2 and significance result from the Mantel test is displayed in the top-left corner.

Effective population size

The Ne estimate for the Pomo Pits group was between 184 –283 depending on the minor allele frequency threshold (Table 2). For the group representing sites outside of Pomo Pits in the Adriatic, the Ne estimate was 131 –144. However, the upper confidence interval for all estimates was infinity, suggesting that, even with samples pooled to regional levels, sample sizes should be increased in future studies to ensure reliable estimates of Ne (Marandel et al., 2020).

Table 2 Estimates of effective population size (Ne) in the Adriatic Sea.

Ne was calculated using the linkage disequilibrium method with random mating assumed, and jackknife 95% confidence intervals were computed.

	Pomo Pits	Outside Pomo Pits	
N e MAF0.01	283 (70–Inf)	131 (54–Inf)	
N e MAF0.02	195 (58–Inf)	141 (53–Inf)	
N e MAF0.05	184 (60–Inf)	144 (54–Inf)	
Notes.

Pomo Pits includes the following sites: Pom1, Pom2, Pom3.

Outside Pomo Pits includes the following sites: 17I, 18II, Anc, Cgg.

Discussion

Pomo Pits and GSA 17

The carapace length results from our study closely match the findings of Angelini et al. (2020), that is, females are on average smaller than males, and both females and males in Pomo Pits are on average much smaller than animals outside of Pomo Pits. Our genetic results indicate that Nephrops from all sites sampled in the Adriatic Sea have high gene flow between them. Likewise, using 890 samples from 27 locations genotyped at 730 SNPs, a recent Mediterranean-wide study of Nephrops did not detect any genetic differentiation within regions, including within the Adriatic (Spedicato et al., 2022: pages 493–513). This suggests that the phenotypic differences attributed to the Pomo Pits subpopulation, namely smaller mean sizes, slower growth rates and smaller mean onset at first maturity (SOM) (Angelini et al., 2020), are not explained by random genetic drift or a lack of gene flow between neighbouring Adriatic subpopulations. Moreover, Pomo Pits Nephrops have no apparent differences in genetic diversity compared to Nephrops at surrounding sites. A recent Scientific, Technical and Economic Committee for Fisheries (STECF) report showed that average Nephrops biomass from 1994-2020 is much higher in Pomo Pits (2,912) compared with Ancona (806) and GSA 18 (1,187) and outlined that Ancona and GSA 18 are at relatively lower biomass levels compared to Pomo/Jabuka Pits (Scientific Technical Economic Committee for Fisheries, 2023), both points of which accord with our genetic diversity estimations.

Our finding of regional gene flow is not unique to Nephrops across other parts of their range, nor to other closely related decapod species of the Adriatic. Using 14 neutral microsatellite markers, Pavičić et al. (2020) found the European lobster (Homarus gammarus) exhibits panmixia and comparable measures of genetic diversity in sites sampled across the Adriatic. These findings suggest that, as with Nephrops in our study, gene flow (and/or high effective population sizes) is likely sufficient to mitigate genetic drift for both species across the sites sampled. However, connectivity modelling of Nephrops has revealed the potential presence of three subpopulations in the Adriatic (Melaku Canu et al., 2021), including one along the northern Croatian coast for which no samples were available for genetic analysis in our study. This would suggest that connectivity in the region likely follows a stepping-stone model, resulting in putative genetic homogeneity across the entire Adriatic region that is caused by adequate ocean current movement, and a sufficiently long pelagic duration (1–2 months; Dickey-Collas et al., 2000) for Nephrops larval dispersal to ensure widespread admixture. Indeed, even where biophysical models strongly overestimate the realised dispersal of typical larvae, a small subset of distantly-dispersing individuals can still drive genetic homogeneity across expansive regions by inducing sufficient gene flow to offset genetic drift and effectively nullify differentiation between otherwise self-recruiting populations (Shanks, 2009; Macleod et al., 2024). Genetic connectivity at similar spatial scales has also been reported for Nephrops across Scandinavia in the north-east Atlantic, for which the authors likewise conclude that patterns of ocean currents and connectivity via larval drift are likely responsible for the observed genetic homogeneity (Westgaard, Søvik & Johansen, 2023). Although we found no genetic structuring in our sites sampled across the Adriatic, additional sampling in the eastern Adriatic will be needed to confirm panmixia of Nephrops across the entire Adriatic Sea.

In light of these findings, what could have driven the observed biological differences of Nephrops in Pomo Pits if they are not caused by genetic isolation? Other lobsters have demonstrated marked variation in SOM between discrete geographic stocks that are nevertheless highly connected genetically. Mean SOM for H. gammarus was estimated to be 18 mm CL larger for males and 31 mm CL larger for females in western Scotland than it was in eastern Scotland (Lizarraga-Cubedo et al., 2003), despite the stocks showing minimal genetic differences (Jenkins et al., 2019; Ellis et al., 2023), while the mean SOM of southern rock lobsters (Jasus edwardsii) declined markedly from 112 mm CL to just 59 mm CL along a latitudinal gradient in Tasmania (Gardner et al., 2006), despite regional genetic homogeneity (Villacorta-Rath et al., 2022). Indeed, the slower growth rates evidenced by Pomo Pits Nephrops may drive their smaller mean sizes and SOM compared to equivalent individuals in surrounding areas (Angelini et al., 2020); maturation and fecundity are often driven by age rather than size in lobsters (Gardner et al., 2006; Ellis et al., 2015), so slower growth reduces mean sizes and renders individuals smaller at a given age, reducing SOM. Environmental factors, especially temperature, are known to influence growth and SOM. Pomo Pits are considerably deeper than surrounding Adriatic areas we sampled (Table 1) and, at ∼10 °C, seawater in the Pomo depressions is typically cooler than shallower surrounding benthos (Artegiani et al., 1993). Yet, while the effect of lower temperatures on metabolism may explain the slower growth of Nephrops in Pomo Pits, this relationship is typically inversed in benthic ectotherms when it comes to SOM; for American lobsters (Homarus americanus), reduced SOM is associated with increased temperatures (Le Bris et al., 2017). Historic fishing effort may be another possible driver of lower SOM among the Nephrops of Pomo Pits. Haarr et al. (2017) modelled potential explanatory factors against spatial variation and temporal declines in the SOM of H. americanus, and, while temperature and population density were poorly correlated to SOM variations, intensity of size-based harvest selection was significantly associated, suggesting that SOM decreases were an evolutionary response to fishing pressure. Similarly, recent results from the Western Mediterranean suggest a sharp decline of Nephrops SOM which appears to be linked to increasing overexploitation of stocks (Vigo et al., 2024). However, although Nephrops stocks in the Adriatic are among the most heavily fished in the Mediterranean (Russo et al., 2019), it is not possible to assess whether spatial differences in harvest intensity may be linked to the lower SOM of Pomo Pits Nephrops; fine-scale spatial data on historic fishing effort have been lacking until recent advances in vessel tracking, and data from contemporary years suggests that other Adriatic stocks we sampled (i.e., Anc and 18II) are subject to similar levels of pressure (Russo et al., 2018). Therefore, although we cannot attribute it to any genetic divergence, any combination of (i) differences in fishing pressures over time and space, (ii) local environmental conditions, and/or (iii) intraspecific competition for space and food could have played a part in driving the exhibited variation in SOM of Pomo Pits Nephrops (Russo et al., 2018; Angelini et al., 2020; Chiarini et al., 2022; Vigo et al., 2024).

Intra-Mediterranean homogeneity

We sampled two Mediterranean sites outside of the Adriatic Sea, one in the Aegean Sea (Aeg) and one in the Tyrrhenian Sea (9I); both sites showed genetic similarity with each other and with the Adriatic sites. Our finding of minimal genetic divergence between Adriatic Nephrops and samples from adjacent sub-basins to the East and West reflects the pattern of population structure observed for the European spiny lobster (Palinurus elephas), for which samples from the Croatian Adriatic showed minimal differentiation to those from either Crete or the Balearic Sea via neutral SNPs (Ellis et al., 2023). However, H. gammarus has a similar estimated window of pelagic larval duration (PLD) to Nephrops, but does show Mediterranean sub-structuring, with Adriatic samples differentiated from those of both the Aegean Sea and the Western Mediterranean (Pavičić et al., 2020; Ellis et al., 2023). Although our depiction of homogeneity between Nephrops of the Adriatic and adjacent areas may in part reflect a paucity of samples from these sites and other regions, our analytical methods and sample sizes for Adriatic Nephrops may be sufficient to explore connectivity at this scale, since previous research has detected even subtle regional differentiation with relatively low sample sizes using 6,340 RAD-derived SNPs (e.g., Ellis et al., 2023).

Atlantic-Mediterranean divergence

At a basin-wide scale, the site sampled in the Atlantic (Firth of Clyde) was strongly differentiated from all Mediterranean sites (pairwise Fst values: 0.11 –0.13). Although only seven individuals were genotyped from this site, this pattern of Atlantic-Mediterranean divergence has also been found in previous studies of Nephrops using mitochondrial D-loop variation (Gallagher et al., 2018) and microsatellite genotypes (Gallagher et al., 2022), though homogeneity was reported using restriction fragment length polymorphisms (Stamatis et al., 2004) and allozymes (Stamatis et al., 2006), which is likely due to the lower power and resolution of these latter two molecular markers to detect genetic variation.

A pronounced phylogenetic break between Atlantic and Mediterranean populations is a characteristic common to a multitude of taxa of the north-eastern Atlantic, despite often greatly varying life history strategies (Patarnello, Volckaert & Castilho, 2007), including littoral fishes (Galarza et al., 2009), benthic echinoderms (Carreras et al., 2020) and bivalve molluscs (Wenne et al., 2022). More relevantly, other wide-ranging lobsters of the region have also demonstrated hierarchical population structure, the primary feature of which is Atlantic-Mediterranean differentiation; European lobster (H. gammarus) and spiny lobster (P. elehpas) both show strong divergence across an Atlantic-Mediterranean divide (Jenkins et al., 2019; Ellis et al., 2023). This concordant pattern amongst different lobster species using putatively neutral SNP loci suggests that, at this scale, Atlantic and Mediterranean meta-populations of all three species have been sufficiently isolated for long enough to develop differing allele frequencies and accumulate new neutral mutations through genetic drift. The drivers of this isolation are likely shaped by the topographic and bathymetric enclosure of the Mediterranean, past and present oceanographic barriers, and periodic vicariance during the Pleistocene glaciations (Patarnello, Volckaert & Castilho, 2007; Pascual et al., 2017; Jenkins et al., 2019). All of these factors serve to inhibit dispersal between the Atlantic and Mediterranean, limiting larval exchange and thus gene flow between populations in each basin (Ellis et al., 2023).

Conclusions

Our results show that, using neutral SNP loci, the Nephrops Pomo Pits subpopulation is not genetically different from surrounding sites in the Adriatic Sea, which is likely explained by connectivity and gene flow between them. In addition, we observed strong genetic differentiation between the Atlantic and the Mediterranean, which supports evidence for an Atlantic-Mediterranean phylogenetic break in Nephrops that has also been reported in many other marine species. These findings suggest that other evolutionary mechanisms, such as phenotypic plasticity or adaptation, are driving the phenotypic differences observed in Pomo Pits Nephrops, which could be linked to differences in fishing pressure over time and space, to local environmental conditions, and/or to intraspecific competition for space and food. The molecular mechanisms underpinning these observed changes in phenotype are most likely linked to many loci under strong selection pressures. As an example, using thousands of individuals and whole genome sequencing, Therkildsen et al. (2019) were able to explain the polygenic mechanisms that underpin rapid evolutionary change of an estuarine fish to smaller body sizes in response to fishing. To explore similar questions in Pomo Pits Nephrops, such an approach, as well as a quality reference genome, improved genomics resources building upon the de novo transcriptome already published (Rotllant et al., 2017), and more comprehensive sample sizes, would be needed.

Supplemental Information

Supplemental Information 1 Results of modelling carapace length as a function of site and sex in Norway lobsters

The authors would like to thank all the people who helped in the field with sample collection, including Irene Moltó Martín, Silvia Angelini, Roberto Cacciamani, Federico Calì, Giovanni Canduci, Matteo Chiarini, Camilla Croci, Filippo Domenichetti and Lorenzo Zacchetti.

Additional Information and Declarations

Competing Interests

Author Contributions

DNA Deposition

Data Availability

The authors declare there are no competing interests.

Tom L. Jenkins conceived and designed the experiments, performed the experiments, analyzed the data, prepared figures and/or tables, authored or reviewed drafts of the article, and approved the final draft.

Michela Martinelli performed the experiments, authored or reviewed drafts of the article, and approved the final draft.

Charlie D. Ellis analyzed the data, authored or reviewed drafts of the article, and approved the final draft.

Jamie R. Stevens conceived and designed the experiments, authored or reviewed drafts of the article, and approved the final draft.

The following information was supplied regarding the deposition of DNA sequences:

The raw DNA sequence data are available from the Sequence Read Archive: PRJNA1100511.

The following information was supplied regarding data availability:

The SNP genotypes in VCF format and R code used to analyse data are available at Zenodo: Tom Jenkins. (2024). Tom-Jenkins/nephrops_rad: Publication Release (v1.0). Zenodo. https://doi.org/10.5281/zenodo.13902611.

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
