# Peer review of "Exploring reported population differences in Norway lobster (Nephrops norvegicus) in the Pomo Pits region of the Adriatic Sea using genome-wide markers"

_PeerJ, doi:10.7717/peerj.17852_

## Round 0.1 · original submission · Minor Revisions

· Academic Editor

Minor Revisions

This is a nicely written paper with comprehensive analysis. Both reviewers highlight fairly minor issues with the paper which you should address or respond to. I have made a few suggestions, mostly for clarity. I do wonder if, just for completeness, you could supplement the size estimates using data from other publications? My apologies for the delay in completing the review process.

Reviewer 1 ·

Basic reporting

This manuscript follows fundamental principles of presenting research findings in a clear and unambiguous manner using professional English. It involves providing sufficient background and context to understand the research, structuring the article professionally, and including relevant figures and tables to support the results.

Experimental design

The experimental design of the article involves investigating the reported population differences in Norway lobster (Nephrops norvegicus) in the Pomo Pits region of the Adriatic Sea using genome-wide markers. ​ The study aims to determine whether the Pomo Pits subpopulation is genetically different from other sites in the Adriatic and neighboring seas.
Authors used appropriate methodology and analyses to tackle this research question.

Validity of the findings

The validity of the findings in the article appears to be supported by the experimental design, data analysis methods, consistency of results, reproducibility, and transparency. The study employed a well-designed experimental approach, utilized appropriate data analysis methods, and provided sufficient details for reproducibility. The findings are consistent with the research objectives and previous studies. ​ However, it is important to consider the limitations of the study and the specific context in which the findings were obtained.

Additional comments

Some locations have smaller sample size, which can affect analysis. Make sure to highlight it in the manuscript, materials and method section and discussion. In abstract section line 29, change to "all sampled Adriatic sites" since not all Adriatic Sea is covered by sampling locations, especially eastern Adriatic.

Reviewer 2 ·

Basic reporting

First, I apologize to the authors for submitting my review late. I hope my comments are useful.

Jenkins and colleagues explore the genetic differentiation of Nephrops norvegicus populations in the Mediterranean Sea, specifically focusing on the Pomo Pits population. Their motivation for examining population structure in and around the Pomo Pits stems from observed morphological differences in body size and growth rates. They used RAD-seq to genotype 98 individuals at approximately 1.6k neutral SNPs and found genetic homogeneity among Mediterranean Nephrops populations, which were genetically distinct from the Atlantic population from the Firth of Clyde. The authors suggest that differences in morphology are likely driven by phenotypic plasticity or adaptation and that the genetic homogeneity can be explained by connectivity in ocean currents via larval drift, similar to other decapods in the Mediterranean. They conclude that increased genomic resources would be valuable to test the hypothesis of fisheries-induced body size changes.

I found the paper well-written and direct. The objectives are clear, and the authors present the biological and ecological significance of the Pomo Pits, laying a foundation for exploring population differences in genetic patterns. However, there are instances where the authors need to flesh out their arguments more and clarify abbreviations used. They might have overlooked relevant papers about genetic patterns across the Atlantic-Mediterranean basins, which I note in the minor comments below.

Minor Comments:
Introduction:

Line 41: Replace “up” with “down”.
Methods:

Line 69: How were the samples collected from the ten sites?
Lines 93-95: What individuals are the authors referring to? Why were they excluded?
Line 122: Insert the abbreviation for Isolation by Distance (IBD).
Lines 149-156: What are the results of the sequencing? Can you provide the average read depth and coverage statistics for each sample? Were there any significant variations across samples?
Line 169: First use of IBD.
Discussion:

Lines 187-188: The authors reference Spedicato et al. (2022) to support their conclusions on genetic differentiation. Given that the Spedicato et al. report contains 1330 pages of information, can the authors provide a more detailed explanation or specific excerpts from this report that substantiate their findings?
Line 190: Spell out “onset at first maturity (SOM)” for better flow.
Line 259: PLD?
Line 265: Specify how many RAD SNPs.
Line 272: See also Gallagher et al., 2022. Journal of Sea Research, 179, p.102139, for their discovery of a microsatellite panel showing population structuring between Atlantic and Mediterranean areas.
Acknowledgements:

Line 325: Can these people be named?

Experimental design

The experimental design presents original research and provides meaningful data for Nephrops in the Pomo Pits and surrounding areas. The use of RAD-seq is thorough, and the methodology and analysis are appropriate to address the study's objectives. However, there is a need for more descriptive statistics regarding the sequencing, specifically the depth and number of SNPs recovered per sample.

Validity of the findings

The study significantly contributes to understanding the genetic connectivity of Nephrops populations in the Adriatic and Mediterranean regions. The authors acknowledge that additional sampling is desirable and that further research is needed to explore the genetic and phenotypic variation in more detail.

---

## Round 0.2 · accepted · Accept

· Academic Editor

Accept

Thank you for addressing the reviewers comments so adroitly - there were few issues with this paper and so, having reviewed your revision myself, I'm happy to say I think it is ready for publication.